# Can Cognitive Neuroscience inform Neuro-Symbolic Models?

**Shashank Srikant, Una-May O'Reilly**
CSAIL, MIT
shash@mit.edu, unamay@csail.mit.edu

## Abstract

The use of neuro-symbolic methods to supplement the performance of deep learning based inference models has witnessed a resurgence. In this work, we review three sets of recent results in human cognition experiments – in natural language comprehension, in natural language inference, and in computer program comprehension - a field bearing similarities to natural language. In light of these three works, we discuss the broader role cognitive neuroscience can play in informing the design of neuro-symbolic model architectures for language.

## 1 Introduction

The use of neuro-symbolic methods to supplement the performance of deep learning based language inference models has witnessed a resurgence [Ilievski *et al.*, 2020; Wang *et al.*, 2020; Ma *et al.*, 2019; Oltramari *et al.*, 2020]. While attempts are constantly being made to improve neural models by incorporating external sources of knowledge to make them more *human-like*, we ask whether our notion of *human-like* performance on language tasks matches our true abilities.

Evidence from vision has shown how our understanding of the biology and neuroscience of vision can directly inform the design of computational models for specific vision-related tasks [Pinto *et al.*, 2009; Woźniak *et al.*, 2020; Shi *et al.*, 2017]. Such evidence for language models is sparse though. In this work, we survey three recent results in human subject studies in language and language-like tasks. Two of these studies investigate brain regions involved in language tasks, and the other investigates behavioral responses to a language inference task. Analyzing results from these studies, we offer two suggestions on using these results to inform and improve the design of neuro-symbolic systems.

We first summarize the three studies by describing their key setup and results, and follow it up with a discussion on the two insights we have to offer (Section 2).

### 1.1 Neural regions involved in language comprehension by Diachek *et. al.*

This study [Diachek *et al.*, 2020] investigates the neural regions involved in language comprehension. The authors investigate the following two brain regions which have been

discovered and established over the last two decades. See [Diachek *et al.*, 2020] for details and references.

**Language system.** These regions have been identified to respond to both comprehension and production of natural language across modalities (written, speech, sign language), respond to typologically diverse languages ($\sim$ 50 languages, from across 10 language families), form a functionally integrated system, reliably and robustly track linguistic stimuli, and have been shown to be causally important for language.

**MD system.** Generally located in the prefrontal and parietal areas of the brain, Multiple Demand system of regions [Duncan, 2010] is known to be domain-agnostic, and is activated in a host of tasks requiring working memory and general problem solving skills, including math and logic.

**Key results.** The authors find that the MD system is not activated for language comprehension tasks, while the language system is consistently activated. Despite language understanding seemingly requiring logic and symbolic manipulation to be applied – cognitive functions generally associated with the MD system, the lack of significant activity in the MD system challenges our intuition of how we cognitively process language. However, these results do not rule out the influence of the MD system in our ability to understand and infer language. The authors suggest that the MD system could likely be recruited for language production tasks, and in comprehension/inference tasks in everyday *noisy channel* conditions.

### 1.2 Human performance on NLI tasks by Pavlick *et. al.*

In the study by [Pavlick and Kwiatkowski, 2019], the authors analyze human performance on inference tasks in language. They study responses to the textual entailment (RTE) task, which expects conclusions to be drawn about the world on the basis of limited information expressed in natural language. For example, the sentence *Three dogs on a sidewalk* being true implies that the sentence *There is more than one dog here* is true. They perform this study on 50 human subjects, wherein each subject is presented 100 such entailment sentence pairs and is required to respond with one of either *entailment*, *neutral*, or *contradiction* for each pair.

**Key results.** The authors show that humans consistently disagree on this task, and report a multi-modal distribution in their responses. Further, and importantly, they find that the

uncertainty expressed by humans is not captured by state of the art inference models like BERT fine-tuned on this RTE task.

## 1.3 Neural regions involved in program comprehension by Ivanova and Srikant *et. al.*

The authors in [Ivanova *et al.*, 2020] studied brain responses to two programming languages – Python and ScratchJr with a goal to understand the regions of the brain involved in comprehending programs.

They disambiguate two cognitive processes likely involved in reading and understanding programs – *code comprehension* – the act of parsing a snippet of source code and understanding the meaning conveyed through the syntax and semantics used in it, and *code simulation* – simulating the parsed program to derive and compute the final output. The latter process mimics the working of an interpreter, and relates to computational thinking, which can be exercised even without programming knowledge [Guzdial, 2008].

The authors investigate two brain regions which may be activated in response to these two cognitive processes – the Language system and the Multiple Demand (MD) system. See Section 1.1 for a brief description of these two regions.

**Key results.** This work establishes that *code comprehension* does not activate our language system and instead consistently activates the MD system. Further, *code simulation* also consistently activates only the MD system.

## 2 The role of Cognitive Neuroscience

The three different results presented above when read together suggest the following observations on how cognitive neuroscience can inform the design of neuro-symbolic systems.

### 2.1 Establishing human performance

The study by Pavlick *et. al.* (discussed in Section 1.2) suggests a partial understanding of our own capabilities and limitations on inference tasks. While recent advances in probing such language models for various properties [Hewitt and Liang, 2019; Voita and Titov, 2020] is a step in the right direction in understanding these models better, they focus primarily on interpreting information learned by these black-box language models. Ambiguity faced by humans during inference is currently neither explained nor modeled in such language models.

There is a need to learn and acknowledge such gaps in our abilities, and use such results to motivate the design of 'general-purpose' language models like BERT and ELMo. It is unclear though how such reconciliation can be operationalized. One observation is that neuro-symbolic systems are trained by integrating external knowledge sources to a learning model. It is possible that with the right choice of external knowledge sources, such as formal logic, relational reasoning, *etc.*, this integration might result in a performance similar to ours.

We see these ambiguities as litmus tests for any system – be it fully neural, or neuro-symbolic, in explaining human-like

cognition. Establishing such ambiguities, and having neuro-symbolic models replicate them will be a worthy initial challenge. Encoding these ambiguities and integrating them in the design of neuro-symbolic models can be another challenge to follow it.

### 2.2 A case for separate architectures

The two studies in the neural bases of language and program comprehension tasks (Sections 1.1, 1.3) provide a different perspective. Despite the assumption that language tasks involve components of logic and general problem solving, Ivanova and Srikant *et. al.* show that they do not activate the MD system system. If we were to design a cognitively inspired computational model, this result may suggest that language tasks are best modeled using an architecture which solely mimics the language system. However, Diachek *et. al.* also suggest that the role of MD system is not well understood for language understanding tasks under noisy conditions, and that the MD system may play a role in such cases. Whether inference tasks, such as common-sense question answering or the RTE task from Pavlick *et. al.*, are treated as noisy conditions is yet to be established. Further, results from Ivanova and Srikant *et. al.* suggest that program comprehension and simulation, which typically entail computational and symbolic manipulation, strongly activate only the MD system and not the language system. These results seem to suggest that there is a need for two distinct architectures to support the broad range of reasoning and inference tasks in language that we engage in – one which models the MD system and the other the language system. We raise the question of whether evidence from how we anatomically process and infer language-related tasks can directly transfer to the design of neuro-symbolic computational models. It is reasonable to conceive a neurosymbolic model comprising two sub-models – one trained exclusively on symbolic and logic-related reasoning, and the other resembling a language model.

## 3 Conclusion

Consolidating results from recent studies in cognitive neuroscience, we offer two insights which promise to deepen our understanding of neuro-symbolic models for language. First, we point the community to studies which establish our own uncertainties when performing language tasks. We believe that neuro-symbolic systems may be better equipped to explain and model these uncertainties. Second, studies on the neural bases of language and computation might suggest the need to explicitly model the 'symbolic' components of any language task. We hope to see these ideas validated in future designs of neuro-symbolic models.

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
