# OpenReview forum: "Can Cognitive Neuroscience inform Neuro-Symbolic Inference Models?"
_ijcai.org/IJCAI/2021/Workshop/NSNLI — NSNLI Oral_

### Official Review · Reviewer_qg4K · 2021-05-25
**Intriguing position paper on neuro-symbolic models**

**Rating:** 7
**Confidence:** 3

**Review:**

I think this paper will generate interesting discussions at the workshop.

The paper surveys three studies from cognitive neuroscience related to the brain's processing of natural language and computer code. The studies mainly focus on the "language system" and the "Multiple Demand" (MD) system of the brain. They show that language comprehension does not rely on MD, which is associated with logical problem solving, while code comprehension and simulation rely solely on MD, and not on the language system.

The paper concludes with recommendations on modeling our architectures for different problems on the language system or the MD system, depending on the task.  The position paper leaves a number of unanswered questions:
- Do we know much about the MD system and the language system to inspire our own architectures?
- How should the two sub-models interact with each other?
- The symbolic models in neuro-symbolic, to my understanding, are assumed to be logic- or program-based. But of course the MD system is neural. Elaborating on what is meant by "symbolic" is important.

---

### Decision · Program_Chairs · 2021-05-27

**Decision:**

Accept (Oral)

**Comment:**

The paper addresses the topic clearly relevant to the workshop.